# Gintonin Alleviates HCl/Ethanol- and Indomethacin-Induced Gastric Ulcers in Mice

**DOI:** 10.3390/ijms242316721

**Published:** 2023-11-24

**Authors:** Han-Sung Cho, Tae Woo Kwon, Ji-Hun Kim, Rami Lee, Chun-Sik Bae, Hyoung-Chun Kim, Jong-Hoon Kim, Sun-Hye Choi, Ik-Hyun Cho, Seung-Yeol Nah

**Affiliations:** 1Ginsentology Research Laboratory and Department of Physiology, College of Veterinary Medicine, Konkuk University, Seoul 05029, Republic of Korea; newlove0202@nate.com (H.-S.C.); bioskjh@konkuk.ac.kr (J.-H.K.);; 2Department of Science in Korean Medicine, Graduate School, Kyung Hee University, Seoul 02447, Republic of Korea; twoo7875@naver.com; 3College of Veterinary Medicine, Chonnam National University, Gwangju 61186, Republic of Korea; 4Neuropsychopharmacology and Toxicology Program, College of Pharmacy, Kangwon National University, Chunchon 24341, Republic of Korea; 5College of Veterinary Medicine, Biosafety Research Institute, Chonbuk National University, Iksan-City 54596, Republic of Korea; 6Department of Animal Health, College of Health and Medical Services, Osan University, Osan-si 18119, Republic of Korea

**Keywords:** gintonin, gastric ulcer, anti-inflammation, prostaglandin E2, LPA5

## Abstract

Gintonin, newly extracted from ginseng, is a glycoprotein that acts as an exogenous lysophosphatidic acid (LPA) receptor ligand. This study aimed to demonstrate the in vivo preventive effects of gintonin on gastric damage. ICR mice were randomly assigned to five groups: a normal group (received saline, 0.1 mL/10 g, p.o.); a control group (administered 0.3 M HCl/ethanol, 0.1 mL/10 g, p.o.) or indomethacin (30 mg/kg, p.o.); gintonin at two different doses (50 mg/kg or 100 mg/kg, p.o.) with either 0.3 M HCl/ethanol or indomethacin; and a positive control (Ranitidine, 40 mg/kg, p.o.). After gastric ulcer induction, the gastric tissue was examined to calculate the ulcer index. The expression of gastric damage markers, such as tumor necrosis factor (TNF)-α, cyclooxygenase 2 (COX-2), and LPA2 and LPA5 receptors, were measured by Western blotting. Interleukin-6 (IL-6) and prostaglandin E2 (PGE2) levels were measured by enzyme-linked immunosorbent assay. The platelet endothelial cell adhesion molecule (PECAM-1), Evans blue, and occludin levels in gastric tissues were measured using immunofluorescence analysis. Both HCl/ethanol- and indomethacin-induced gastric ulcers showed increased TNF-α, IL-6, Evans blue permeation, and PECAM-1, and decreased COX-2, PGE2, occludin, and LPA5 receptor expression levels. However, oral administration of gintonin alleviated the gastric ulcer index induced by HCl/ethanol and indomethacin in a dose-dependent manner. Gintonin suppressed TNF-α and IL-6 expression, but increased COX-2 expression and PGE2 levels in mouse gastric tissues. Gintonin intake also increased LPA5 receptor expression in mouse gastric tissues. These results indicate that gintonin can play a role in gastric protection against gastric damage induced by HCl/ethanol or indomethacin.

## 1. Introduction

The gastric mucosa is a tissue that withstands exposure to gastric acid, digestive enzymes, ingested alcohol, and other food-derived irritants; it also has to endure physical stresses induced by contact with rough solid and high-temperature food boluses [1]. Defense strategies for the gastric mucosa include the regulation of gastric acid secretion, formation of a mucus layer, and bicarbonate secretion into the mucosal spaces. Mucosal blood flow enhancement is an additional means of cyto-protection, as it supplies the nutrients and growth factors required for cell viability and repair of mucosal lesions [2].

Gastric ulcers are among the most common stomach diseases. Gastric acid, pepsin, Helicobacter pylori infection, alcohol consumption, and non-steroidal anti-inflammatory drugs (NSAIDs) are the major causes of gastric ulcers [3]. Alcohol can perturb the gastric mucosa and induce numerous metabolic changes, leading to mucosal damage and stomach lesions [4]. Thus, alcohol is an ulcerative agent that causes mucosal damage when combined with gastric acid, directly damaging cells and increasing gastric mucosal permeability [5]. Moreover, the long-term administration of NSAIDs triggers gastric ulcers by damaging the gastric mucosa [6]. Cyclooxygenase 2 (COX-2) inhibition, which results in the reduction of prostaglandin E2 (PGE2) synthesis, oxidative stress, and increases in inflammation caused by alcohol and NSAIDs, is one of the major causes of gastric pathogenesis [7,8]. In addition, there are increases in inflammatory cytokines after alcohol and NSAIDs [7,8].

Chemosynthetic drugs including H2 receptor blockers, proton pump inhibitors, and gastric mucosal protective agents are currently used to treat gastric ulcers. However, although effective, these drugs tend to cause severe adverse reactions such as hepatitis, diarrhea, and headache [9]. Therefore, several extracts and active components derived from natural herbs have been investigated as alternatives to these drugs [10,11].

Ginseng, the root of *Panax ginseng*, is a well-known traditional herbal medicine in the Far East that has been used as a tonic for many centuries [12,13,14]. Recent studies have demonstrated that ginseng contains gintonin, a non-saponin, and non-polysaccharides [14]. Lysophosphatidic acids (LPAs), which are small bioactive lysophospholipids, are the bioactive components of gintonin [15]. To date, at least six G protein-coupled LPA receptor subtypes have been identified (named LPA1-LPA6), which have various biological functions, such as cell proliferation, survival, and migration in the nervous and non-nervous systems [16].

However, although the gastrointestinal system expresses diverse LPA receptor subtypes, a systematic evaluation of the gastrointestinal protective effects of gintonin, particularly on HCl/ethanol- and NSAID-induced gastric injury animal models, has not been reported [17,18,19]. In this study, since these animal models provide similar symptoms with human gastric damage as described above, we examined the protective effect of gintonin against gastric mucosal damage induced by HCl/ethanol and NSAID via LPA receptor subtype(s). We found that oral administration of gintonin protected against gastric damage induced by HCl/ethanol and NSAID.

## 2. Results

### 2.1. Effects of Gintonin on 0.3 M HCl/Ethanol- or Indomethacin-Induced Gastric Ulcers in Mice

The mice receiving physiological saline did not show any gastric lesions upon macroscopic observation (Figure 1, Control), whereas the mice 0.3 M HCl/ethanol-treated mice group showed severe gastric damage, such as visible bleeding mucosal lesions (Figure 1A,B). Thus, the administration of 0.3 M HCl/ethanol generated 22.4 ± 3.2% of the ulcerated stomach area. The ranitidine group administered orally as a positive control generated 4.8 ± 0.1% of the ulcer area (*p* = 0.006, Cohen’s d = 2.0). The oral administration of different dosages of gintonin (GT 50 or GT 100) generated 9.0 ± 1.9% (*p* = 0.052, Cohen’s d = 4.9) and 3.8 ± 0.9% (*p* = 0.001, Cohen’s d = 6.8) of the ulcer sites, respectively.

We subsequently examined the effects of gintonin on indomethacin-induced gastric ulcers. As shown in Figure 1C,D, indomethacin administration generated 14.2 ± 2.9% of the ulcerated area. However, oral ranitidine, administered as a positive control, generated 3.3 ± 1.1% of the ulcer area. The gintonin 50 or 100 mg/kg group also generated 6.7 ± 0.6% (*p* = 0.06. Cohen’s d = 2.4) and 3.3 ± 0.2% (*p* = 0.004, Cohen’s d = 3.4) of the ulcer areas, respectively. These results show that oral gintonin administration resulted in a significant reduction in both 0.3 M HCl/ethanol- and indomethacin-induced gastric ulcers in a dose-dependent manner. We observed that both gintonin (GT 100)- and ranitidine-treated groups had similar gastric ulcer indices in attenuating gastric ulcers induced by 0.3 M HCl/ethanol and indomethacin (Figure 1B,D).

### 2.2. Effects of Gintonin on the Gastric Morphological Changes Induced by 0.3 M HCl/Ethanol or Indomethacin

H&E staining is widely used for the recognition of various tissue types and morphological changes under normal and abnormal conditions for observing a wide range of cytoplasmic, nuclear, and extracellular matrix features [20]. Administration of 0.3 M HCl/ethanol or indomethacin caused damage to the mucosal layer of the gastric tissue (Figure 2A,B). However, when gintonin or ranitidine was administered before gastric ulcer induction, the damage to the mucosal layer due to gastric ulcers was reduced in a dose-dependent manner (Figure 2A,B). In particular, the gintonin (100 mg/kg, GT 100)-administered group demonstrated a tissue shape similar to that of ranitidine-treated gastric ulcers.

### 2.3. Gintonin Protects against Gastric Vascular Damages Induced by 0.3 M HCl/Ethanol or Indomethacin

Evans blue is a fluorescent dye that specifically binds to albumin in the plasma. Therefore, Evans blue staining is commonly used to assess vascular permeability [21]. In this study, we used vascular endothelial growth factor (VEGF) as a gastric blood vessel marker [22]. When Evans blue was injected into a vein through the orbital plexus, the amount of gastric Evans blue increased in the group treated with 0.3 M HCl/ethanol or indomethacin and gastric blood vessels also were damaged with loss of blood vessel forms (*p* = 0.005, Cohen’s d = 2.0 and *p* = 0.0004, Cohen’s d = 2.5, respectively) (Figure 3A,C). However, the amount of Evans blue decreased in the gintonin treatment group compared to in the 0.3 M HCl/ethanol or indomethacin alone group (*p* = 0.003, Cohen’s d = 2.0 and *p* = 0.003, Cohen’s d = 2.1, respectively) (Figure 3B,D). In particular, the gintonin (GT 100)-treated group showed a trend similar to that of the ranitidine-treated group (*p* = 0.004, Cohen’s d = 2.1 and *p* = 0.024, Cohen’s d = 1.5, respectively), in which the blood vessels were close to those of the normal group. These results indicated that administration of 0.3 M HCl/ethanol or indomethacin caused gastric vascular damage and that gintonin administration protected gastric blood vessels (Figure 3B,D). These results also showed that gintonin administration prevented gastric ulcers and gastric blood vessel damage induced by 0.3 M HCl/ethanol or indomethacin.

### 2.4. Gintonin Alleviates Platelet Endothelial Cell Adhesion Molecule (PECAM-1) and Occludin Changes Induced by 0.3 M HCl/Ethanol and Indomethacin

PECAM-1 and occludin are representative proteins that constitute gap junctions between the gastric blood vessel barrier and gastric mucous layer [23,24]. As described above, Evans blue staining revealed that gintonin administration protected against gastric blood vessel damage. Next, we examined whether the gintonin-induced protection of gastric blood vessels was achieved via gap junction protein protection. The degree of PECAM-1 and occludin expression was visualized by immunofluorescence staining. Gastric ulcers induced by 0.3 M HCl/ethanol or indomethacin increased the level of PECAM-1, whereas the level of occludin decreased in the gastric tissue (Figure 4A,C,E,G). However, gintonin administration attenuated this increase in PECAM-1 immunofluorescence (Figure 4B,D), and increased occludin immunofluorescence staining (Figure 4E,H). These results indicate that gastric ulcers damage the gap junctions of the blood vessels. However, gintonin administration restored the alterations in the gap junction proteins. These results also show that gintonin-mediated gastric blood vessel protection was achieved via gap junction protection.

### 2.5. Effect of Gintonin on the Cyclooxygenase-2 (COX-2) and PGE2 Level in 0.3 M HCl/Ethanol- or Indomethacin-Induced Gastric Ulcers

Cyclooxygenase-2 (COX-2) is involved in the conversion of arachidonic acid to prostaglandin E2 (PGE2), which is expressed during inflammation [25]. However, in the stomach, PGE2 protects the stomach by promoting gastric mucus secretion and decreasing gastric acid secretion [26]. Gastric COX-2 and PGE2 levels are significantly reduced in gastric ulcers [27]. As shown in Figure 5A–F, treatment with 0.3 M HCl/ethanol and indomethacin reduced the COX-2 expression (*p* = 0.022, Cohen’s d = 12.8 and *p* = 0.001, Cohen’s d = 5.1, respectively) and PGE2 levels (*p* = 0.027, Cohen’s d = 5.7 and *p* = 0.000008, Cohen’s d = 3.2, respectively) in the mouse stomach. However, gintonin administration increased gastric COX-2 expression (*p* = 0.039, Cohen’s d = 4.9 and *p* = 0.004, Cohen’s d = 6.8, respectively) and PGE2 level in the 0.3 M HCl/ethanol- and indomethacin-treated groups (*p* = 0.029, Cohen’s d = 13 for 0.3 M HCl/ethanol test group; *p* = 0.00018, Cohen’s d = 1.4 for gintonin 50 mg/kg and *p* = 0.0000001, Cohen’s d = 4.6 for gintonin 100 mg/kg for indomethacin test group, respectively). In particular, the gintonin (100 mg/kg)-treated group showed COX-2 expression and PGE2 levels similar to those of the ranitidine-treated group.

### 2.6. Gintonin Increases the Secretion of Gastric Mucus in the Stomach Damaged by 0.3 M HCl/Ethanol and Indomethacin

Since gintonin administration increased COX2 expression and PGE2 levels in the stomach under insults of 0.3 M HCl/ethanol and indomethacin, we subsequently examined the effects of gintonin on the gastric mucous after 0.3 M HCl/ethanol and indomethacin treatment. Alcian blue is a dye that specifically binds to sugars without binding to lipids or proteins. In particular, Alcian blue pH 2.5 solution is commonly used to measure acidic mucus [28]. We found that mice treated with 0.3 M HCl/ethanol (*p* = 0.0029, Cohen’s d = 2.5) and indomethacin (*p* = 0.015, Cohen’s d = 1.5) demonstrated a significant loss of gastric mucus content compared to that of the normal group. However, treatment with gintonin (*p* = 0.0003 for 0.3 M HCl/ethanol group, Cohen’s d = 5.25 and *p* = 0.011, Cohen’s d = 3.4 for Indomethacin group) or ranitidine (*p* = 0.012, Cohen’s d = 2 for 0.3 M HCl/ethanol group and *p* = 0.018, Cohen’s d = 2.4 for Indomethacin group) prevented the significant loss of gastric mucus (Figure 5G,H). These results show that gintonin administration can protect the stomach wall under insults of 0.3 M HCl/ethanol and indomethacin by increasing mucous secretion in the stomach wall.

### 2.7. Effect of Gintonin on IL-6 and TNF-α Expressions in the Stomach Damaged by 0.3 M HCl/Ethanol- and Indomethacin

IL-6 and TNF-α are cytokines and representative cell-signaling proteins associated with inflammatory reactions and increase during inflammatory reactions. Elevated IL-6 and TNF-α levels are known to be closely related to the development of various immune abnormalities, inflammatory diseases, and lymphatic tumors [29]. Therefore, we subsequently checked whether gintonin administration can attenuate inflammatory reactions under insults of 0.3 M HCl/ethanol and indomethacin; we examined the effects of gintonin on the expression level of IL-6 and TNF-α after 0.3 M HCl/ethanol and indomethacin treatment. As shown in Figure 6A,B, treatment of 0.3 M HCl/ethanol or indomethacin significantly increased IL-6 (*p* = 0.000009, Cohen’s d = 2.6 and 0.0000009, Cohen’s d = 7.6, respectively) and TNF-α (*p* = 0.002, Cohen’s d = 8.2 and 0.004, Cohen’s d = 4.6, respectively) in the ulcer group in the stomach tissues compared to the normal group (N). However, gintonin and ranitidine treatment attenuated the amount of IL-6 expression by 24 ± 3.8% with 50 mg/kg gintonin (*p* = 0.0015, Cohen’s d = 1.4 and 0.043, Cohen’s d = 1.3, respectively), 74 ± 12.6% with 100 mg/kg gintonin (*p* = 0.000003, Cohen’s d = 4.7 and *p* = 0.000009, Cohen’s d = 6.2, respectively), and 75 ± 9.5% with ranitidine (*p* = 0.000004, Cohen’s d = 5.1 and *p* = 0.000001, Cohen’s d = 6.7, respectively).

Similarly, gintonin or ranitidine treatment attenuated the amount of TNF-α expression by 17 ± 6.1% in gintonin 50 mg/kg (*p* = 0.05, Cohen’s d = 3 and 0.07, Cohen’s d = 1.7 respectively), 67 ± 1.5% in gintonin 100 mg/kg (*p* = 0.0005, Cohen’s d = 4.5 and 0.0002, Cohen’s d = 6.4, respectively), and 70 ± 3.2% in the ranitidine group (*p* = 0.0008, Cohen’s d = 3.9 and *p* = 0.0006, Cohen’s d = 7.5, respectively). These results indicate that 0.3 M HCl/ethanol or indomethacin induces gastric inflammation and that gintonin-mediated gastric protection is also involved in the anti-inflammatory actions.

### 2.8. Gintonin Induces the Expression of the LPA5 Receptor

There are six LPA receptor subtypes, and LPA2 and LPA5 receptor subtypes are expressed in the gastrointestinal mucous membranes [30,31]. Gintonin is an exogenous ligand of lysophosphatidic acid (LPA) receptors [32]. To determine whether the LPA2 and LPA5 receptors are involved in gastric ulcers induced by 0.3 M HCl/ethanol and indomethacin, we first examined the LPA2 and LPA5 receptor subtype expression levels under insults of 0.3 M HCl/ethanol and indomethacin by western blotting. In the ulcer group, the expression level of the LPA5 subtype was significantly decreased upon treatment with 0.3 M HCl/ethanol (*p* = 0.0003, Cohen’s d = 5.6) and indomethacin (*p* = 0.015, Cohen’s d = 2.6); however, the LPA2 receptor subtype was not affected under 0.3 M HCl/ethanol and indomethacin (Appendix A). Moreover, treatment with gintonin restored the depleted LPA5 receptor subtype expression (*p* = 0.0007, Cohen’s d = 4.4 for gintonin 50 mg/kg and *p* = 0.003, Cohen’s d = 5 for gintonin 100 mg/kg in 0.3 M HCl/ethanol test group, respectively; *p* = 0.022, Cohen’s d = 2.0 for gintonin 50 mg/kg and *p* = 0.045, Cohen’s d = 1.4 for gintonin 100 mg/kg in Indomethacin test group). These results suggest that gastric ulcers induced by 0.3 M HCl/ethanol and indomethacin in mice decreased LPA5 receptor expression, but gintonin treatment restored LPA5 receptor expression and alleviated the gastric ulcers induced by 0.3 M HCl/ethanol and indomethacin (Figure 7).

## 3. Discussion

The stomach is a hollow, muscular organ in the gastrointestinal tract of both animals and humans. The stomach has a dilated structure and functions as a vital digestive organ [33]. Acute and chronic alcohol intake induces severe gastric mucosal damage in humans and rodents and rapidly penetrates the gastric mucosa, causing membrane damage, cell exfoliation, and erosion. The subsequent increase in mucosal permeability together with the release of vasoactive products from mast cells, macrophages, and other blood cells can lead to vascular injury, necrosis, inflammation, and ulcer formation [34].

NSAIDs have been used clinically as anti-inflammatory and analgesic agents. However, ulcer induction in the gastrointestinal tract is a major side effect of these drugs and a major limitation of their use [35,36]. For example, indomethacin, a non-corticosteroid drug, induces ulcerative gastric damage. These agents also cause lipid peroxidation in the mucosa and play important roles in the development of gastric mucosal lesions [36].

In contrast, gintonin, newly isolated from ginseng, consists of glucose, proteins, and lipid complexes, such as glycolipoproteins, and acts as an exogenous LPA receptor ligand that activates LPA receptor subtypes [37]. LPAs are present in various body fluids and organs including saliva [38]. LPAs are also present in the stomach for gastric protection [19], and the main roles of LPAs are cell proliferation stimulation, survival, and migration via six LPA receptor subtypes [16]. However, little is known about the protective effects of gintonin against the effects of HCl/ethanol and NSAIDs. In the present study, we examined whether gintonin alleviates gastric ulcers induced by HCl/ethanol or NSAIDS with possible involvement of LPA receptor subtypes in mice, as acid/ethanol-induced gastric ulcer animal models are commonly used to investigate the pathogenesis and develop therapies for human ulcerative diseases [39]. Therefore, we examined the effects of gintonin on HCl/ethanol- and NSAID-induced ulceration in mice.

Gintonin administration attenuated HCl/ethanol- and indomethacin-induced macroscopic gastric ulcerative lesions in mice in a dose-dependent manner (Figure 1). However, oral administration of gintonin itself in tested dosage had no effects on normal mouse stomach. Microscopic histological studies demonstrated that gintonin administration alleviated the HCl/ethanol- and indomethacin-induced microscopic gastric mucosal layer damage in a dose-dependent manner (Figure 2). These results showed that gintonin protected the stomach from gastric ulcers induced by HCl/ethanol or indomethacin. In the molecular mechanism studies of how gintonin protects against gastric damage from both HCl/ethanol and indomethacin, we found that gintonin mediates protection against gastric damage through the following mechanisms.

First, gintonin administration attenuated gastric blood vessel damage induced by HCl/ethanol and indomethacin. As shown in Figure 3, gintonin administration attenuated the blood vessel- and gap junction-related protein changes induced by HCl/ethanol and indomethacin. Thus, gintonin administration attenuated PECAM-1 expression, which increases when gap junctions are damaged. However, gintonin administration increased occludin expression, which decreases when gap junctions are damaged. Moreover, gintonin administration restored gastric permeability-related elements, as shown by Evans blue permeability experiments (Figure 3 and Figure 4). The protective effects of gintonin on gastric blood vessels and gap junction proteins have been linked to the restoration of gastric permeability. These results show that gintonin administration restores gastric damages induced by HCl/ethanol and indomethacin via gastric blood vessel-related gap junction protection.

Second, gintonin administration protected the stomach from COX-2 and PGE2 reduction induced by HCl/ethanol and indomethacin administration. As shown in Figure 5, gintonin administration increased the COX-2 expression and PGE2 levels in mice exposed to HCl/ethanol and indomethacin. HCl/ethanol and indomethacin administration induces gastric ulcers via the inhibition of COX-2 expression and subsequent reduction of gastric PGE2, which is a protective factor that stimulates gastric mucus production. Thus, the effects of gintonin on COX-2 and PGE2 against HCl/ethanol and indomethacin were coupled to increase gastric mucus production, as shown by the gastric Alcian blue test (Figure 5).

Third, gintonin administration attenuated HCl/ethanol- and indomethacin-induced gastric inflammation by inhibiting inflammatory cytokine expression in the stomach tissue. HCl/ethanol and indomethacin are known to induce gastric inflammation by damage to the gastric wall [40]. However, gintonin administration inhibited the overexpression of inflammatory cytokines, such as IL-6 and TNF-α, which are representative inflammation-related cytokines, as shown in Figure 6. Lastly, we examined whether the administration of HCl/ethanol and indomethacin affects gastric LPA2 and LPA5 receptor subtype expression and found that the expression of gastric LPA5, but not the LPA2 receptor subtype, was reduced under HCl/ethanol and indomethacin insults (Figure 7 and Appendix A). These results suggest that HCl/ethanol and indomethacin selectively affect the expression of the gastric LPA5 receptor subtype. However, gintonin administration restored gastric LPA5 receptor subtype expression, indicating that the LPA5 receptor subtype may play a role in gintonin-mediated gastric protection against HCl/ethanol- and indomethacin-induced gastric damage.

In summary, we found that gintonin administration alleviated in vivo HCl/ethanol- and indomethacin-induced gastric damage in both macroscopic and microscopic manners. In the mechanism studies of gintonin-mediated gastric protection against HCl/ethanol and indomethacin, we further found that gintonin restored the decreased level of gastric COX-2 expression and PGE2 levels under HCl/ethanol- and indomethacin-induced gastric damages. Thus, the restored PGE2 protects and restores the gastric mucosal layer, blood vessels, and gap junctions, whereas gintonin administration also attenuates gastric inflammation. Finally, the present study demonstrates the possible role of gintonin as a gastric protecting agent against alcohol intake and NSAIDS. Gintonin can be used as a natural functional food or medicine for stomach health that protects the stomach damaged by alcohol and anti-inflammatory drugs in humans, although the human application of gintonin for stomach health might further require approval through clinical tests.

## 4. Materials and Methods

### 4.1. Gintonin Preparation from Ginseng Root

Gintonin was prepared from *P. ginseng* by using a previously described method [13]. Briefly, 1 kg of 4-year-old ginseng was ground into small pieces (>3 mm) and refluxed with 70% fermentation ethanol eight times for 8 h at 80 °C each. The ethanol extracts (350 g) were concentrated. Ethanol extract was dissolved in distilled cold water at a ratio of 1–10 and stored in a cold chamber at 4 °C for 24 h. The supernatant and precipitate produced by water fractionation, after the ethanol extraction of ginseng, was separated by centrifuge (3000 rpm, 20 min). The precipitate was lyophilized after being centrifuged. This fraction was designated GEF and had a yield of 1.3%. GEF contains about 0.2% LPA C18:2, 0.06% LPA C16:0, and 0.02% LPA C18:1. The chemical was identified using an Agilent series 1100 HPLC (Agilent Technologies, Santa Clara, CA, USA) system with components including a G1311A quart pump, G1313A autosampler, G1322A degasser, G1316A column oven, and an API 2000 LC-MS/MS system (Applied Biosystems, Foster City, CA, USA) [41].

### 4.2. Animal

The experimental animals used in this study were 8-week-old ICR male mice purchased from Koatech (Pyeongtaek, Republic of Korea). The mice were allowed to fully acclimatize for one week before use in the experiment. Mice weighing 30 g were selected and used. The breeding environment had a temperature of 25–27 °C and 50–60% humidity, providing 12 h of light and dark (9:00~21:00). Sufficient water and feed were supplied, and Purina (made in Republic of Korea) was provided as feed. This study was conducted after receiving animal experimentation ethics approval from the Animal Experimentation Ethics Committee of Konkuk University (Deliberation No. KU21051-1). Each experimental group contains at least five mice, respectively.

### 4.3. Gastric Ulcer Induction by HCl/Ethanol or Indomethacin in Mice

An acute gastric hemorrhagic lesion was induced by oral administration (0.1 mL/10 g) of an HCl/ethanol mixture containing 0.3 M HCl in 60% ethanol to ICR (8-week-old male mice). Gintonin was dissolved in saline and administered at an oral dose of 50 or 100 mg/kg 1 h before HCl/ethanol administration. As a positive control, ranitidine, a drug for the treatment of gastric ulcers, was dissolved in saline and administered orally at a dose of 40 mg/kg, 1 h before the application of HCl/ethanol. After 1 h, an HCl/ethanol mixture containing 0.3 M HCl in ethanol was orally administered (0.1 mL/10 g) to induce acute gastric lesions [40]. One hour after HCl/ethanol administration, the animals were sacrificed by cervical dislocation, and the gastric tissue was excised, opened along a larger curvature, and washed with 1× phosphate-buffered saline (PBS) solution. Gastric lesions were induced by oral administration of indomethacin (30 mg/kg) (Thermo-Fisher Scientific, Waltham, MA, USA) to ICR mice (8-week-old male mice). One hour before gastric ulcer induction with indomethacin, gintonin was resuspended in 0.5% CMC (carboxymethylcellulose distilled water solution) at concentrations of 50 and 100 mg/kg. Ranitidine (40 mg/kg), a drug used to treat gastric ulcers, was used as a positive control. After 1 h, indomethacin was administered to induce gastric ulceration. Twelve hours after indomethacin administration, the animals were sacrificed by cervical dislocation, and gastric tissue was excised, opened along the larger curvature, and washed with 1×PBS solution. We performed experiments with each designated dose based on the doses tested in-lab data, with modifications of those tested below. HCl/ethanol (0.1 mL/10 g) mixture was used according to the literature [40], and indomethacin [35,36] and ranitidine were used based on the literature, with modification [40]. The gintonin doses we selected were calculated, based on Nair et al.’s study for possible translation in humans and vice versa [42]. As indicated in the research conducted by Lee et al., the administration of 300 mg per 60 kg of body weight to human subjects and 600 mg per 60 kg of body weight to human subjects resulted in effectiveness without any observed toxicity [43]. In this regard, the doses for human subjects were converted into possible mouse translation.

### 4.4. Evaluation of the Gross Lesions and Gastric Ulcer Area in the Gastric Ulcer

To calculate the gastric ulcer area, measurement of ulcerated areas was applied in a blinded manner using Adobe Photoshop software (Ver. 25.1).

### 4.5. Preparation of the Tissue Samples

Anesthetized mice (8-week-old males) were fixed to a fixture plate. The skin and peritoneum of the mice (8-week-old males) were cut using scissors and opened. A 26-G needle was inserted into the heart apex of the mice and perfused with 1×PBS solution for 5 min. The perfusion system BT100-2J (Longer pump, Shanghai, China) was used at 36 rpm. The needle was removed from the heart after 5 min. Gastric tissue was excised, opened along the larger curvature, and washed with 1×PBS solution. The gastric tissue was washed with 1×PBS, spread on a fixed plate, and fixed using a needle. The fixing plate was immersed in 4% paraformaldehyde (Biosesang, Seongnam, Republic of Korea) and reacted at 4 °C for 48 h to fix it. The fixed tissues were processed as described by Lee et al. (2017) [44].

### 4.6. Hematoxylin and Eosin (H&E) Tissue Staining

The tissue specimens were dipped in xylene three times for 5 min each to remove paraffin. The samples were soaked in 100% ethanol for 5 min and then in 50% ethanol for 5 min to remove the xylene. Nuclei were stained by soaking for 2 min in hematoxylin dye. The immersion was repeated three times with a 0.3% ammonia solution and washed under running water. The sections were incubated with an eosin solution for 20 s to stain the remainder, except for the nucleus. The H&E Staining Kit (Abcam, Cambridge, MA, USA; ab245880) was used in this study. Images of the stained sections were captured using a DP70 digital light microscope (Olympus, Tokyo, Japan).

### 4.7. Determination of the Gastric Wall Mucosal Barrier

The drugs (ranitidine 40 mg/kg, GT 50 or 100 mg/kg) were administered 1 h before gastric ulcer induction. After induction of gastric ulcer from 0.3 M HCl/ethanol for 1 h and indomethacin at 30 mg/kg for 12 h, ICR mice (8-week-old male mice) were sacrificed by cervical dislocation. The gastric tissue was excised, opened along the larger curvature, and washed with a cold sucrose (0.25 M) solution dissolved in distilled water. The tissue was weighed, placed in 10 mL of Alcian blue (0.1%) solution (Sigma-Aldrich Korea, Gangnam, Republic of Korea), and incubated at room temperature for 2 h. After 2 h, the solution was mixed with an equal volume of diethyl ether and centrifuged at 3000 rpm for 15 min. The solution obtained by centrifugation (100 μL) was added to a 96-well plate to read the absorbance at 605 nm using a microplate reader (Molecular Devices, San Jose, CA, USA). The Alcian blue content was calculated using a standard curve of Alcian blue (0, 0.1, 0.3, 1,3, and 10 μg/mL), and the results were expressed in μg of Alcian blue/g of tissue.

### 4.8. Determinations of PGE2 and Interleukin-6 (IL-6) Levels

PGE2 and IL-6 levels were measured in mice with gastric ulceration. One hour before gastric ulcer induction, the drugs (ranitidine 40 mg/kg, GT 50 or 100 mg/kg) were orally administered to the mice. After 1 h, gastric ulceration (0.3 M HCl/ethanol 0.1 mL/10 g for 1 h, indomethacin 30 mg/kg for 12 h) was induced. Heart perfusion was performed using 1×PBS in mice to which gastric ulcers were induced. Measurements were made using a PGE2 enzyme-linked immunosorbent assay (ELISA) Kit (Cat. No. MBS266212) or Mouse IL-6 ELISA Kit (Cat. No. MBS730957) (MYBioSource, San Diego, CA, USA) according to the manufacturer’s instructions. The absorbance was measured at 450 nm using a microplate reader (Molecular Devices, San Jose, CA, USA).

### 4.9. Western Blot Analysis

After inducing gastric ulcer with 0.3 M HCl/ethanol or indomethacin, the mouse was anesthetized with urethane and cardiac perfusion was performed using 1×PBS. The excised gastric tissues were homogenized on ice using scissors. The 2 mL tube containing homogenized tissue was centrifuged at 13,000 rpm and 4 °C for 20 min to obtain a precipitate. After centrifugation, 100 μL of Radioimmunoprecipitation assay (RIPA) lysis buffer (150 mM Sodium Chloride, 0.25% Sodium deoxycholate, 1 mM EGTA, 1% NP-40, 50 mM Tris-HCl pH 8.0) was added to the precipitate and reacted on ice for 1 h. One hour later, a 2 mL tube containing homogenized tissue was centrifuged at 13,000 rpm and 4 °C for 20 min. The supernatant obtained after centrifugation was subjected to protein quantification using the BCA protein kit (Thermo-Fisher Scientific Korea, Gangnam-gu, Seoul, Republic of Korea). All the samples were analyzed after protein quantification. Samples were further processed according to Lee et al. (2017) [44] using COX-2 antibody (1:1000, Cat. No. 160106, Cayman, MI, USA) or tumor necrosis factor (TNF)-α antibody (1:1000, Cat. No. PA5-19810. Thermo Scientific, Waltham, MA, USA).

### 4.10. Immunofluorescence Staining

As mentioned earlier, the gastric ulcer was induced by 0.3 M HCl/ethanol or indomethacin, and prepared gastric tissue sections were used. We removed the paraffin from the tissue sections attached to the slides. The slides were placed on a rack and the next wash was performed: xylene 3 min 2 times→100% ethanol and xylene 1:1 mixture 3 min→100% ethanol 3 min→95% ethanol 3 min→70% ethanol 3 min→50% ethanol 3 min→wash with cold 1×PBS. We added Tris-EDTA buffer (10 mM Tris Base, 1 mM EDTA solution, 0.05% Tween 20, pH 9.0) to the vessel containing the slide rack and incubated overnight in a 60 °C water bath. The next day, we washed the slides twice for 5 min with gentle agitation of TBS and 0.025% Triton X-100. Incubated for 2 h with the addition of 1% BSA (dissolved in TBS) at room temperature to block non-specific binding. After 2 h, 1% BSA was removed using a tissue. Tissue sections (n = 4 per stomach, 4 stomachs per group) were incubated with rabbit anti-vascular endothelial growth factor-A (VEGF-A) (1:500; Abcam, Cambridge, MA, USA), and rat antiplatelet endothelium, cell adhesion molecule, PECAM-1 (1:500; Santa Cruz Biotechnology, Santa Cruz, CA, USA) and mouse anti-occludin (1:500; Invitrogen, Waltham, MA, USA). The humid box is closed carefully by its lid and kept overnight at 4 °C on a flat balanced surface without agitation. The next day, we rinsed with 2 × 5 min TBS 0.025% Triton with gentle agitation. Cyanine 3 (Cy3) and fluorescein isothiocyanate (FITC)-conjugated immunoglobulin G (IgG) antibodies (1:500; Jackson ImmunoResearch, West Grove, PA, USA) as secondary antibodies. We applied fluorophore-conjugated secondary antibody to the slide diluted to the concentration recommended by the manufacturer in TBS with 1% BSA and incubated for 1 h at room temperature. After 1 h, we washed 3 times with 1×PBS for 5 min. Place the coverslip on the tissue section and circle the edges of the coverslip with clear nail polish to prevent the cells from drying out. We set up a negative control to ensure binding to the specific target protein. Images from each slice were captured capitalizing confocal imaging system (LSM 5 PASCAL; Carl Zeiss, Oberkochen, Germany) and their intensity was quantified. DAPI (4′,6-diamidino-2-phenylindole) (1:1000; Thermo-Fisher Scientific, Waltham, MA, USA) was used as counterstaining.

### 4.11. Determination of Gastric Vascular Permeability

The vascular permeability was measured as Evans blue concentration in stomach tissue. The respiratory anesthesia through Isoflurane was performed prior to the administration of Evans blue (Sigma-Aldrich Korea, Gangnam, Republic of Korea). After the administration of drugs (ranitidine 40 mg/kg, GT 50 or 100 mg/kg), 200 μL of Evans blue (0.5%) dissolved in distilled water was administered to the ICR mouse (8-week-old male) through the orbital plexus. 0.3 M HCl/ethanol (0.1 mL/10 g) or indomethacin (30 mg/kg) induced gastric ulcer progressed 1 h later. After a reaction time (0.3 M HCl/ethanol for 1 h and Indomethacin 30 mg/kg for 12 h) the animals were sacrificed by cervical dislocation. The gastric tissue was excised and opened along a larger curvature and washed with 1×PBS solution. The tissues were weighed and incubated with 2.5 mL of formamide (Sigma-Aldrich Korea, Gangnam, Republic of Korea) at room temperature for 24 h. The sample was centrifuged at 3000 rpm and 4 °C for 15 min to obtain supernatant. Samples and standard (Evans blue dissolved in distilled water) were added 100 μL to the 96-well plate to read absorbance at 612 nm using a microplate reader (Molecular Devices, San Jose, CA, USA). The Evans blue content was calculated using a standard curve of Evans blue (0, 0.1, 0.3, 1, 3, and 10 μg/mL), and the results were expressed in μg of Evans blue/g of tissue.

### 4.12. Statistical Analysis

The experiments were repeated at least thrice. Data are expressed as mean ± standard error of the mean. Statistical analysis was performed using the one-way ANOVA followed by the Dunnett/or Tukey test. The Shapiro–Wilk test was used before each statistical analysis to assess the normal distribution. Statistical significance was set at *p* < 0.05. For statistical analysis and diagram creation, SigmaPlot for Windows (version 12.5) was used (Systat Software, Inc., Point Richmond, CA, USA).

## Figures and Tables

**Figure 1 ijms-24-16721-f001:**
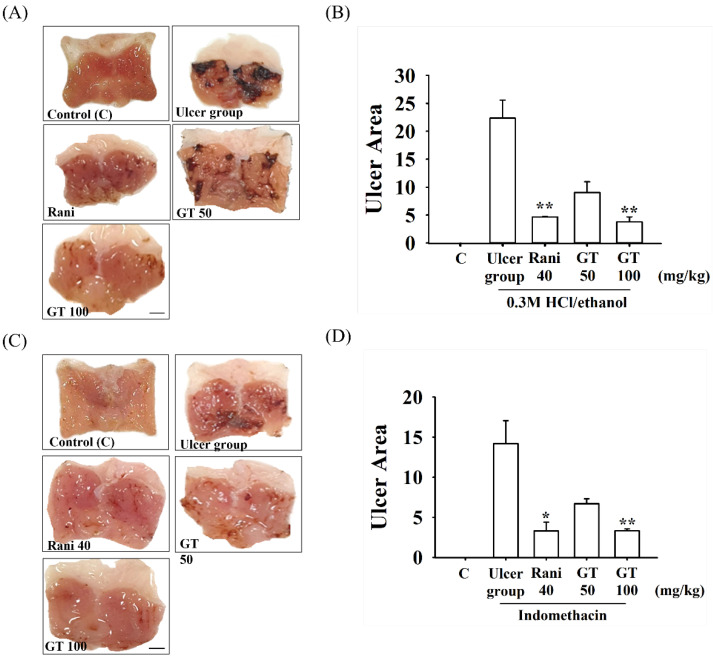
Effects of gintonin on 0.3 M HCl/ethanol- and indomethacin-induced gastric ulcer in mice. A representative photograph of gastric tissue after oral administration of saline control, 0.3 M HCl/ethanol (**A**) or indomethacin (**C**). An hour before 0.3 M HCl/ethanol was administered, gintonin 50 (GT 50) or 100 mg/kg (GT 100) or ranitidine (40 mg/kg) (Rani40) were administered orally. One hour after 0.3 M HCl/ethanol or indomethacin administration, gastric tissue was excised to determine the ulceration index by 0.3 M HCl/ethanol (**A**,**B**) or indomethacin (**C**,**D**). Gintonin or ranitidine administration significantly relieved gastric ulcers in the stomach tissue. Data are expressed as the mean ± standard error of the mean (n = 5). * *p* < 0.05 compared to the ulcer group. ** *p* < 0.01 compared to the ulcer group. Scale bar = 2.5 mm.

**Figure 2 ijms-24-16721-f002:**
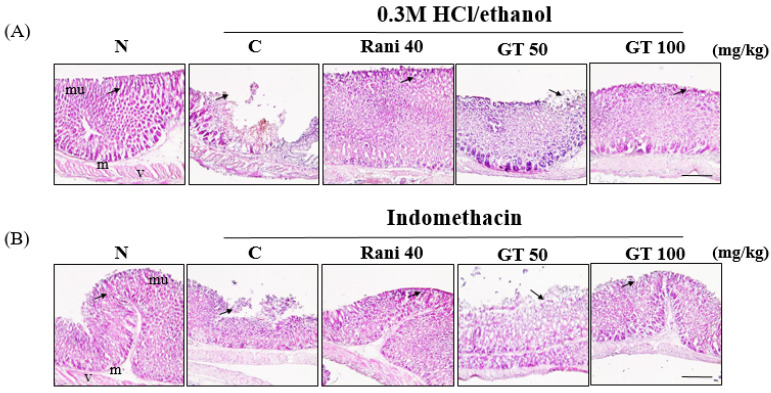
Gintonin attenuates gastric mucosal layer damage induced by 0.3 M HCl/ethanol and indomethacin in mice. (**A**) Gintonin or ranitidine effects on 0.3 M HCl/ethanol-induced gastric mucosal layer damages. (**B**) Gintonin or ranitidine effects on indomethacin-induced gastric mucosal layer damages. Histological structures were marked for the gastric mucosal (mu) and submucosal (m) layers with normal blood vessels (v). 0.3 M HCl/ethanol and indomethacin groups showed mucosal damage due to gastric ulcer (arrow). Gintonin or ranitidine administration before induction of gastric ulcer was shown to reduce gastric mucosal damage (arrow). Scale bar = 100 µm.

**Figure 3 ijms-24-16721-f003:**
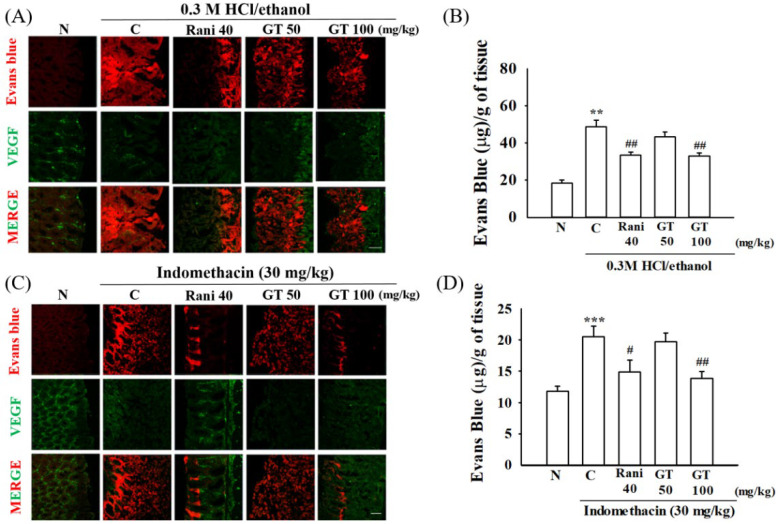
Gintonin reduces vascular leakages induced by 0.3 M HCl/ethanol and indomethacin in mice. (**A**,**C**) Evans blue staining (red color) of gastric tissue was used to determine the degree of blood vessel leakages. VGEF (green color) was used as a counterstaining for blood vessels. The detailed experimental protocols are described in Figure 1 legend. (**B**,**D**) Quantitative histograms on of Evans blue amount in gastric tissue after orbital administration of Evans blue after 0.3 M HCl/ethanol or indomethacin. Scale bar = 100 µm. The data are expressed as the mean ± SEM (n = 5). ** *p* < 0.01 and *** *p* < 0.001 compared to the normal group; ^#^ *p* < 0.05 and ^##^ *p* < 0.01 compared to the control group. MERGE (red and green color); Merge the channel of Evans blue staining and VEGF.

**Figure 4 ijms-24-16721-f004:**
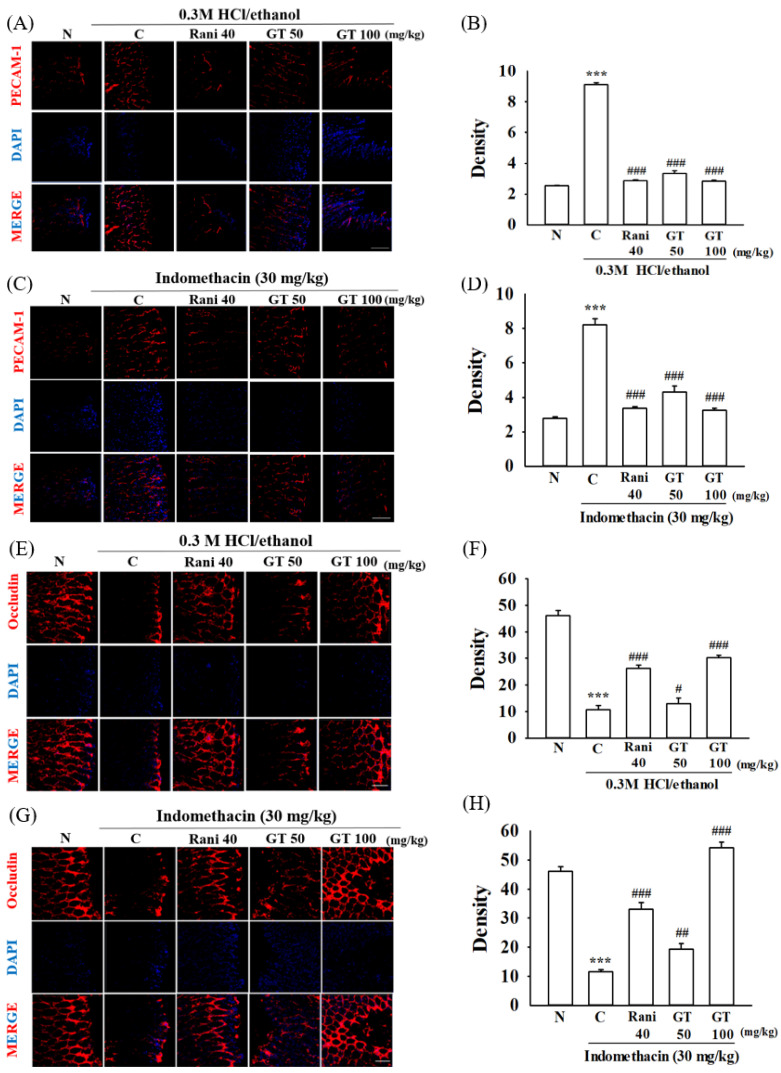
Effects of gintonin on PECAM-1 and occludin protein expressions in gastric ulcer induced by 0.3 M HCl/ethanol and indomethacin ((**A**,**C**,**I**,**K**), 0.3 M HCl/ethanol and (**E**,**G**,**J**,**L**), indomethacin). The degree of expression of PECAM-1 and occludin in gastric tissues was visually confirmed through immunofluorescence staining. Gintonin administration before gastric ulcer induction decreased or increased the level of PECAM-1 (red color) and occludin (red color) in gastric tissue, respectively. DAPI (blue color) was used as a counter staining ((**B**,**D**), 0.3 M HCl/ethanol and (**F**,**H**), indomethacin). The summary histograms of quantifying the density of PECAM-1 and occludin. Scale bar = 100 µm. The data are expressed as the mean ± SEM (n = 5). *** *p* < 0.001 compared to the normal group and ** *p* < 0.01 compared to the normal group; ^#^ *p* < 0.05 and ^##^ *p* < 0.01 and ^###^ *p* < 0.001 and compared to the control group.

**Figure 5 ijms-24-16721-f005:**
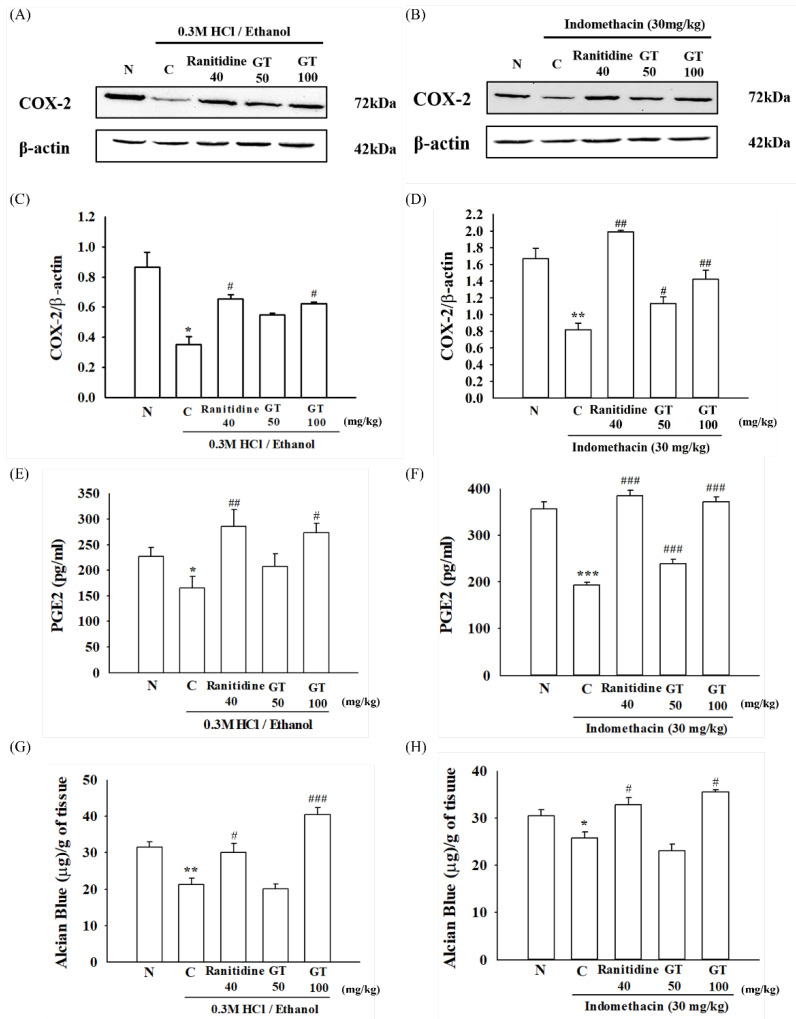
Effect of gintonin on the cyclooxygenase-2 (COX-2) expression, prostaglandin E2 (PGE2) amount, and gastric mucus in gastric ulcer-induced mice induced by 0.3 M HCl/ethanol and indomethacin. The COX-2 enzyme expression was detected by Western blot analysis using β-actin as a control (**A**–**D**) and PGE2 was quantitated using PGE2 enzyme-linked immunosorbent assay (ELISA) kit (**E**,**F**). COX-2/β-actin ratio at each dose and PGE2 amount are summarized (**C**,**D**). (**G**,**H**) The amount of gastric Alcian blue was quantitated after administering gintonin (50 or 100 mg/kg). The detailed experimental protocols are described in the legend for Figure 1. Data are expressed as the mean ± standard error of the mean (n = 5). * *p* < 0.05 and ** *p* < 0.01 and *** *p* < 0.001 compared to the normal group; ^#^
*p* < 0.05 and ^##^
*p* < 0.01 and ^###^
*p* < 0.001 compared to the control group.

**Figure 6 ijms-24-16721-f006:**
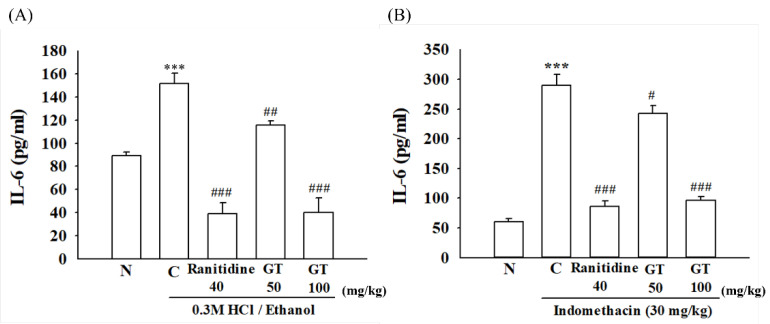
Effect of Gintonin on the interleukin (IL-)6 amount and tumor necrosis factor (TNF)-α expression in gastric ulcer-induced mice. The amount of IL-6 (**A**,**B**) and TNF-α expression level (**C**,**D**) was measured using the Mouse IL-6 enzyme-linked immunosorbent assay (ELISA) kit (**A**,**B**) and western blotting analysis, respectively (**C**–**F**). The detailed experimental protocols are described in the legend for Figure 1. Data are expressed as the mean ± standard error of the mean (n = 5). ** *p* < 0.01 and *** *p* < 0.001 compared to the normal group; ^#^
*p* < 0.05 and ^##^
*p* < 0.01 and ^###^
*p* < 0.001 compared to the control group.

**Figure 7 ijms-24-16721-f007:**
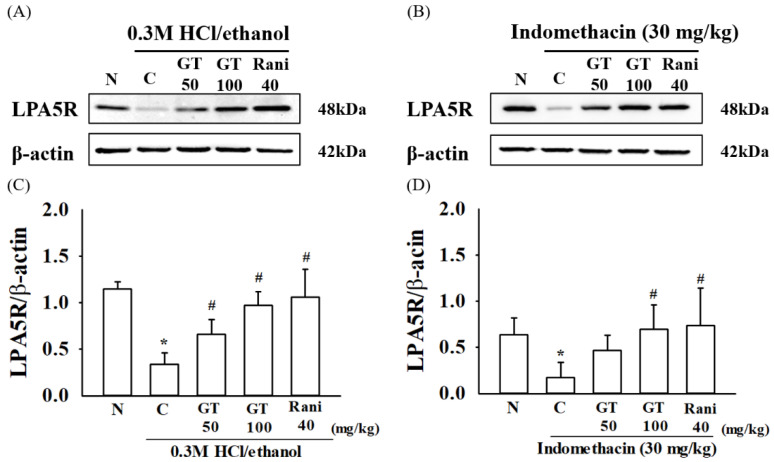
Effect of gintonin on lysophosphatidic acid (LPA)5 receptor expression in gastric ulcer-induced mice. The expression of gastric LPA5 receptor (LPA5R) protein was detected by Western blot analysis. (**A**) Mice were treated with gintonin (50 mg/kg, GT 50), gintonin (100 mg/kg, GT 100) or ranitidine (40 mg/kg) for 1 h and then treated with 0.3 M HCl/ethanol for 1 h. (**B**) Mice were treated with gintonin (50 mg/kg), gintonin (100 mg/kg) or ranitidine (40 mg/kg) for 1 h and then treated with indomethacin for 12 h. (**C**,**D**) The summary histograms on Western blotting as a LPA5R/β-actin ratio at each dose. The data are expressed as the mean ± standard error of mean (n = 5). * *p* < 0.05 compared to the normal group; ^#^
*p* < 0.05 compared to the control group.

## Data Availability

The data presented in this study are available on request from the corresponding author. The data are not publicly available due to privacy limitations.

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
