# Peer review of "Gintonin Alleviates HCl/Ethanol- and Indomethacin-Induced Gastric Ulcers in Mice"

_ijms, 2023, doi:10.3390/ijms242316721_

Round 1

Reviewer 1 Report

Comments and Suggestions for Authors

The manuscript entitled “Gintonin alleviates HCl/ethanol- and indomethacin-induced

gastric ulcer in mice” addresses the beneficial effects of gintonin, an extract of ginseng root in dampening gastric ulcer triggered by ethanol and indomethacin, and some associated mechanisms. Initially, the authors proved that gintonin lowered gastric ulcer area and suppressed gastric mucosal damage. Then, the authors proceeded to some implicated mechanisms. To this end, the authors demonstrated that gintonin curtailed gastric inflammation as evidenced by lowering the levels of TNF-α, IL-6, and PECAM-1. In tandem, gintonin dampened gastric vascular permeability and VEGF expression. Meanwhile, gintonin augmented the expression of the gap junction protein occludin, mucus production, and prostaglandin E2 levels. Finally, the authors proved that gintonin upregulated the expression of lysophosphatidic acid receptor subtype 2 (LPA2) without affecting LPA5. The current findings are interesting.

Comments:     

1) Since gintonin is not a single compound and is, in fact, an extract of the ginseng root, the authors need to show evidence of its chemical characterization. Hence, the authors are advised to add all the chemical charts (e.g., HPLC, NMR, ….) that confirm the identity of isolated compounds in gintonin extract. These data are essential to substantiate the reliability of the present study.

2) In section 4.3., how was the dose of gintonin selected? Is the selected dose in mice relevant for human translation? Please, discuss the dose used for possible translation in humans, for example, by using conversion tables available in the literature using the Human effective dose (HED) formula= animal dose x animal Km/ human Km (Nair AB, Jacob S. A simple practice guide for dose conversion between animals and humans. J Basic Clin Pharm. 2016 Mar;7(2):27-31). Authors are advised to address this point and add the answers/proper citations to this section.

3) Please, add proper citation for the used doses of HCl/ethanol, indomethacin, and ranitidine.

4) Is the dose of gintonin safe in mice? Check the literature to see if toxicological data are available. Alternatively, try to provide early safety data.

5) In the experimental design, why did not the authors incorporate an additional group (control + gintonin 100 mg/kg)? This group may reveal any potential toxicity of the tested agent.

6) In Figure 1, what do the authors specifically mean by ulcer index %? In fact, the data listed by the authors should be described as “ulcer area” rather than ulcer index which is a totally different parameter. In section 4.4., please, add the title “gastric ulcer area”.

7) In immunofluorescence, did the authors also perform a negative control to ensure the specific binding of antibody to target protein? Please, add the answer to the material and methods section.

8) In the statistical analysis section, did the authors check data normal distribution before proceeding to one-way ANOVA? Authors are advised to address this point and add the answers in the material and methods section.

9) Since the study involves several experimental groups/treatments, statistical analysis is typically analyzed by ANOVA followed by a post-hoc test e.g., Tukey-Kramer. The use of the t-test may not be appropriate. The authors are advised to redo the statistical analysis as described.

10) In Figure 1, to avoid readers' confusion, please add the macroscopic image of the normal control (and the ulcer area in panels B and D; as zero). To avoid readers’ confusion, the authors are advised to rename the control as “ulcer group”. Likewise, why did not the authors show the control + gintonin group?

11) The authors are advised to carefully revise the results section and the description of the current findings. For example, in lines 94-95, the authors state “However, oral ranitidine, administered as a positive control, reduced the gastric ulcer area by 3.3 ± 1.1%”. The above statement is not accurate and should be replaced by “However, oral ranitidine, administered as a positive control, reduced the gastric ulcer area to 3.3 ± 1.1%”. This issue needs to be carefully addressed in the entire results section.

12) In supplementary Figure 1, the authors are advised to quantify VEGF protein expression and move it to the main manuscript.

13) How do the authors justify that PECAM1 was increased in the ulcer group (named as control by the authors)? Please, add an explanation in the discussion section and elaborate on PECAM1’s role in leukocyte transmigration during the inflammatory response.

14) The manuscript needs to be carefully checked by a native English speaker for grammar and typos. Some typos/syntax errors are present in the manuscript which need to be addressed, for example:

- In lines 366-367, the authors state “The stomachs were . measurements were performed in a blinded manner using Adobe Photoshop software”.

Please, consider correcting the above statement to  “To calculate the gastric ulcer area, measurement of ulcerated areas was applied in a blinded manner using Adobe Photoshop software”.

15) The authors are advised to carefully revise the reference section. The authors are advised to unify the way they write the journal name. Sometimes it is written as a full name (references 6 and 7) while in other references it was written as an abbreviation (remaining references). Please, follow the journal instructions in this regard.

Minor points:

1) In the introduction section, the authors are advised to elaborate on the 2 experimental animal models for gastric ulcer (HCl/ethanol- and indomethacin-induced gastric ulcer models). Please, describe in detail the main advantage of each model and why it was used in the current study.

2) In line 359, the authors state “gintonin was dissolved in 0.5% CMC (carboxymethylcellulose distilled water solution)”. In fact, CMC is a suspending agent, so, the authors are advised to double-check whether gintonin was dissolved or suspended in CMC.

3) In section 4.3., kindly, provide the number of animals used in each experimental group.

4) The authors are advised to add catalog no. for the used kits and antibodies.

5) Why did the authors add the methodology of gastric vascular permeability and immunofluorescence staining in the supplementary file? Please, add these data to the manuscript.

6) Likewise, please, transfer the figures listed in the supplementary file to the manuscript.  

7) Since gintonin is not a single compound and is in fact an extract of ginseng root, please revise the statement “Gintonin, newly isolated from ginseng, is a glycoprotein that acts as an exogenous lysophosphatidic acid (LPA) receptor ligand”.

Comments on the Quality of English Language

Moderate editing of the English language is required.

Reviewer 2 Report

Comments and Suggestions for Authors

The authors have prepared a paper on the anti-ulcerative properties of ginseng demonstrated on a murine model. Although the topic is not novel, the authors used a thorough methodology and support their findings with illustrations and relevant discussions. There are, however, several improvements to be made before the paper can be considered for publication.

1. Please rephrase the description and identification of the study lots in the abstract; in the current form it is not clear how the lots are composed.

2. more information needs to be inserted in the introduction to explain the why the authors chose to perform the study, what are the unresolved problems regarding the gastro-protective effects of ginseng, and what would be the added contribution of the paper.

3. it is recommended that the authors present the objectives of the study and the hypothesis.

4. please present the p-values in the manuscript in the Results section; add the exact value not only <0.05 so the readers get a better idea of the significance of difference.

5. please consider including Cohen's d exact size in order for the readers to be able to understand the magnitude of the effect

6. please discuss what is the toxic dose of gintonin and whether the effective doses reported in this study exceed the maximum acceptable amount.

7. please name and describe the statistical software used.

8. please add a Data Statement regarding the availability of the data used, whether it can be found in data repositories or how the readers can access further details on the measurements

9. please add a paragraph addressing the limitations of the study in the latter part of the Discussions section

10. it would be beneficial to discuss why indomethacin was chosen for this particular animal model and the particular properties that would be translatable to human studies

Respectfully submitted,

Comments on the Quality of English Language

The language is generally fine, with the exception of several small grammar and phrasing issues

Round 2

Reviewer 1 Report

Comments and Suggestions for Authors

The authors addressed most of the raised comments. Thanks.

Comments on the Quality of English Language

Minor editing of the English language is required.